# Does the Position of Foot-Mounted IMU Sensors Influence the Accuracy of Spatio-Temporal Parameters in Endurance Running?

**DOI:** 10.3390/s20195705

**Published:** 2020-10-07

**Authors:** Markus Zrenner, Arne Küderle, Nils Roth, Ulf Jensen, Burkhard Dümler, Bjoern M. Eskofier

**Affiliations:** 1Machine Learning and Data Analytics Lab, Department of Computer Science, Friedrich-Alexander-Universität Erlangen-Nürnberg (FAU), 91052 Erlangen, Germany; arne.kuederle@fau.de (A.K.); nils.roth@fau.de (N.R.); bjoern.eskofier@fau.de (B.M.E.); 2Finance & IT - IT Innovation, Adidas AG, 91074 Herzogenaurach, Germany; ulf.jensen@adidas.com (U.J.); burkhard.duemler@adidas.com (B.D.)

**Keywords:** wearable computing, foot kinematics, sensor position, zero velocity update, inertial measurement unit, sport science, running

## Abstract

Wearable sensor technology already has a great impact on the endurance running community. Smartwatches and heart rate monitors are heavily used to evaluate runners’ performance and monitor their training progress. Additionally, foot-mounted inertial measurement units (IMUs) have drawn the attention of sport scientists due to the possibility to monitor biomechanically relevant spatio-temporal parameters outside the lab in real-world environments. Researchers developed and investigated algorithms to extract various features using IMU data of different sensor positions on the foot. In this work, we evaluate whether the sensor position of IMUs mounted to running shoes has an impact on the accuracy of different spatio-temporal parameters. We compare both the raw data of the IMUs at different sensor positions as well as the accuracy of six endurance running-related parameters. We contribute a study with 29 subjects wearing running shoes equipped with four IMUs on both the left and the right shoes and a motion capture system as ground truth. The results show that the IMUs measure different raw data depending on their position on the foot and that the accuracy of the spatio-temporal parameters depends on the sensor position. We recommend to integrate IMU sensors in a cavity in the sole of a running shoe under the foot’s arch, because the raw data of this sensor position is best suitable for the reconstruction of the foot trajectory during a stride.

## 1. Introduction

Wearables have become increasingly important in many fields of our everyday life. Among applications in medicine, workplaces, and many others, the sports domain was one of the early adopters of wearable technology. The reasons for the quick spread of small body-worn sensors in sports was due to the manifold advantages of the technology for athletes, researchers, and the sports industry. Using wearables, athletes can utilize low-cost sensor technologies in order to enhance their performance, prevent injuries, and improve their motivation [1]. Sports research benefits from the fact that wearables allow in field data acquisitions, whereas many studies in the sports domain were traditionally laboratory bound [2]. Furthermore, the sports industry can, on the one hand, offer innovative and more attractive sports products with integrated sensor technology, and, on the other hand, gather consumer data which they can use to improve their products.

One sport where wearable technology already has a great impact is endurance running. Both professional and recreational runners track themselves and use online platforms like Runtastic (Runtastic GmBH, Pasching, Austria) or Strava (Strava, San Francisco, CA, USA) to monitor their training progress and performance. Three heavily used sensor technologies in endurance running are Global Positioning System (GPS) trackers, heart rate monitors, and inertial measurement units (IMUs). While GPS trackers like smartwatches or smartphones are utilized for visualizing the running track and providing real-time feedback on pace and distance [3], heart rate monitors are used to evaluate the physical effort of runs and provide real-time feedback on exercise intensity and training effect [4]. However, these two sensor modalities are not capable of revealing insights into the biomechanics of runners. IMUs are low-cost sensors which consist of 3D-accelerometers measuring linear acceleration as well as 3D-gyroscopes measuring angular velocity. By attaching those sensors to different parts of the human body, various endurance running-related biomechanical parameters can be computed and evaluated.

In this context, IMUs can be used in various ways. Researchers developed body sensor networks with multiple sensors in order to reconstruct the movement of different extremities in a synchronized manner [5]. The advantage of those sensor networks is the holistic evaluation of runners’ movements. However, attaching all the sensors requires a lot of time which recreational runners are often not willing to spend for everyday runs. That is why researchers also investigated single sensors at specific positions on the human body which can easily and quickly be attached. Apart from placing sensors on the lower back [6,7], the tibia [8,9], or the ankle [10], a popular sensor position used in literature is the foot or the running shoe [2,11,12,13,14,15]. One reason for the popularity of this sensor position is the amount of different spatio-temporal parameters that can be computed. Falbriard et al. [16] showed that foot-mounted IMUs can be used to accurately segment running strides into their sub-phases (ground contact phase and swing phase) and thus allow computing stride time and ground contact time. Apart from that, researchers developed algorithms to reconstruct the trajectory, that is, orientation and translation, of the foot during a stride. A popular approach for the computation of the trajectory is strapdown integration using a zero-velocity assumption during midstance [17]. From the resulting foot translation, stride length and average stride velocity was calculated with a mean error of 2 cm and 0.03 m/s, respectively [15]. Additionally, the orientation can be used to compute angular foot kinematic parameters like the sole angle in the sagittal plane or the range of motion in the frontal plane [12,14].

However, publications using foot-mounted IMUs differ not only in the computed spatio-temporal parameters, but also in the position of the IMU sensors on the running shoes. Shiang et al. [11], Falbriard et al. [12], and Strohrmann et al. [2] mounted the IMU sensors on the instep of the foot on top of the shoelaces, whereas Lederer et al. [13] and Koska et al. [14] mounted the sensors on the heel. Other sensor positions presented in literature are on the lateral side of the running shoe below the ankle [17] and inside the sole of the running shoe [18,19].

To our knowledge, the effect of the IMU sensor position on the foot with respect to raw data quality and accuracy of spatio-temporal parameters has not been evaluated yet. Peruzzi et al. [20] evaluated the best possible IMU sensor position for algorithms using zero-velocity updates and found a lateral mounting below the ankle to be the best sensor location. They evaluated the quality of the sensor position using a motion capture system by investigating the motion of the retroreflective markers at different locations. However, they did not include IMU raw data or spatio-temporal parameters to evaluate the sensor position.

We contribute an in-depth analysis by comparing different sensor positions with respect to the raw signals as well as their suitability for the computation of spatio-temporal parameters. We base our evaluation on a study with 29 subjects, which wore running shoes with IMUs attached to the instep, the heel, the lateral side of the foot, and within the cavity of the running shoe. We compare the similarity of IMU raw data at different sensor positions by computing Pearson’s correlation coefficients between the individual sensor positions’ raw data. Besides, we compute and evaluate temporal stride features based on state-of-the-art event detection algorithms and spatial features using a zero-velocity-based strapdown integration algorithm.

## 2. Methods

### 2.1. Definition of Spatio-Temporal Parameters

The temporal parameters we evaluated were stride time $t_{stride}$ and ground contact time $t_{gc}$ (Figure 1). One stride was defined by two consecutive initial ground contacts (ICs) of the same foot. The duration between those time instances is the stride time. One stride could be further segmented into ground contact phase and swing phase by finding the toe off (TO) event where the foot leaves the ground. The duration of the ground contact phase is called ground contact time. One further important phase during ground contact time is midstance (MS). It subdivides the ground contact phase into absorption and propulsion phase. We used the MS for the zero-velocity update in the strapdown integration algorithm.

The spatial parameters based on the translation of the foot were stride length $d_{stride}$ and the average stride velocity $v_{stride}$. The stride length is defined as the translation of the foot during one stride. The average stride velocity can be computed by dividing the stride length by stride time.

We used the sole angle and the range of motion as spatial parameters based on the orientation to evaluate the sensor positions (Figure 2). The sole angle is defined as the angle between the sole of the running shoe and the ground in the sagittal plane at IC. The range of motion describes the eversion movement of the foot during ground contact. Runners land on the lateral side of their foot and rotate inwards after IC. The angle describing the amount of inward rotation is the range of motion in the frontal plane.

### 2.2. Data Set

We collected data of 29 amateur runners (23 male; 6 female) with a mean age of 24.9±2.4 years. All subjects were informed about related risks and gave written consent to participate in the study and for the collected data to be published. The data was acquired in a laboratory with a motion capture system (Vicon Motion Systems Inc., Oxford, UK) as reference. All subjects wore the same kind of running shoes (adidas Response Cushion 21, Adidas AG, Herzogenaurach, Germany). Both the left and the right shoe were equipped with four IMU sensors. The sensors were located in a cavity in the sole of the running shoe, laterally under the ankle, at the heel, and on the instep (Table 1; Figure 3).

For the study, we used miPod IMU sensors [21]. The accelerations $\vec{a}\left[n\right]$ and angular rates $\vec{\omega }\left[n\right]$ at sample *n* measured with those sensors will be denoted as
(1)$$\vec{a}\left[n\right]=\left(\begin{array}{c} a_{x}\left[n\right]\\ a_{y}\left[n\right]\\ a_{z}\left[n\right] \end{array}\right)\;\mathrm{and}\;\vec{\omega }\left[n\right]=\left(\begin{array}{c} \omega_{x}\left[n\right]\\ \omega_{y}\left[n\right]\\ \omega_{z}\left[n\right] \end{array}\right)\;,$$
where indices *x*, *y*, and *z* denote the vector components along the axis of the respective sensor (Figure 3). The sensors sampled accelerations and angular rates with a frequency of $f_{s}=200$ Hz and a resolution of 16 bit. According to Potter et al. [22], we set the range of the accelerometer to ±16 g and the range of the gyroscope to $\pm 2000^{\circ }$/s. Prior to the data acquisition, the miPod sensors were calibrated using the calibration routine introduced by Ferraris et al. [23]. Additionally, a functional calibration routine was performed to align the individual sensors’ coordinate systems with the shoe coordinate system $\left(x_{s},y_{s},z_{s}\right)$. The functional calibration routine (Figure 4) was performed for each subject and generated two vector pairs for each sensor. Each vector pair consisted of one vector in the sensor frame and one vector in the shoe frame:(1)Vector pair superior/inferior direction: The subjects were asked to stand still with both feet on the ground. Thus, the accelerometer of all sensors measured the gravitational acceleration in the sensor frame. The $z_{s}$-axis was defined as the corresponding vector in the shoe frame.(2)Vector pair medial/lateral direction: The subjects rotated their feet on a balance board, which only allowed for a rotation in the shoe frame’s sagittal plane. A gyroscope in the shoe frame measures the angular rate of the rotation on the medial/lateral axis. The medial/lateral axis of the shoe frame corresponds to the principle component of the angular rate data during rotation in the sensor frame. The $x_{s}$-axis was defined as the medial/lateral axis in the shoe frame.

Using these vector pairs, we computed a subject dependent rotation matrix for each sensor location, which rotated the IMU-data in the sensor frame into the shoe frame using an adapted version of the Whaba algorithm [24,25]. After applying this rotation matrix to the sensor data, all sensor frames were aligned with the shoe frame, which makes the raw IMU data comparable on each axis. Besides, the functional calibration offers the possibility to run the same algorithms on each sensor due to the same alignment of the sensors and thus enable a fair comparison between the sensor positions. For the simplicity of the notation in this work, we keep the convention for acceleration $\vec{a}\left[n\right]$ and gyroscope $\vec{\omega }\left[n\right]$ defined in Equation Equation 1, but use *x*, *y*, and *z* as the axis in the shoe coordinate system from now on.

The motion capture ground truth system consisted of 16 infrared cameras and sampled the positional data of the retroreflective markers with a sampling rate of $f_{s}=200$ Hz. The running shoes were equipped with a subset of the marker setup described by Michel et al. [26]. For our study, we only used the six markers attached to each foot. Using these markers and the marker-based stride segmentation method for IC and TO using motion capture data introduced by Maiwald et al. [27], the reference values for the spatio-temporal stride features could be estimated.

The sensors and the motion capture system were synchronized using an adapted version of the wireless trigger introduced by Kugler et al. [28]. Due to small differences in the IMUs sampling rates, this procedure only allowed for a stride-to-stride synchronization, not a sample-to-sample synchronization.

In the described set-up, each subject was asked to run 50 times through the motion capture volume. We controlled for speed by capturing different number of trials in different velocity ranges of 2–6 m/s (Table 2). Using the described study set-up, we were able to collect data of 2426 strides.

An example stride for the four IMU sensors, which was segmented from IC to IC, is depicted in Figure 5.

### 2.3. Algorithm

#### 2.3.1. Stride Segmentation

We used a combination of different existing stride event detection algorithms to find IC, MS, and TO in the IMU signal. IC and TO were computed by finding maxima in the angular rate data of the sagittal plane. According to Falbriard et al. [16], these two maxima are reliable features to detect IC and TO even though they have a certain bias from the actual IC and TO.

We found a two stage approach to be most reliable for IC detection. Firstly, we used a cyclicity estimator of Šprager et al. [29] to find the swing phase index $n_{SP,i}$ before the *i*-th ground contact at the local minimum of the gyroscope in the sagittal plane (Figure 6). This instant in time corresponds to the forward swing of the foot during swing phase. From this fiducial point, we searched for the next local maximum of the gyroscope signal in the sagittal plane, which corresponds to the point of maximum rotation in the sagittal plane after $n_{SP,i}$. Falbriard et al. [16] reported a bias of 11 ms for this fiducial point to the actual IC event. Due to this fact and the chosen sampling rate of 200 Hz, we corrected the detected maximum by two samples to find the index of the *i*-th IC $n_{IC,i}$.

From this time instance, the index of the *i*-th MS $n_{MS,i}$ was determined by finding the minimum in the gyroscope L2-norm after $n_{IC,i}$ [30] in a time range of 250 ms (50 sample), which is the average time of a stance phase while running with speeds up to 6 m/s [31]:(2)$$n_{MS,i}=\underset{n}{\mathrm{argmin}}\sqrt{\omega_{x}^{2}\left[n\right]+\omega_{y}^{2}\left[n\right]+\omega_{z}^{2}\left[n\right]}\;,n\in \left[n_{IC,i};n_{IC,i}+50\right]$$

For the detection of TO, we used the first maximum in the angular rate data after the maximum related to IC in a time range of 400 ms [31]. We added 150 ms to the average stance time for speeds up to to 6 m/s to account for running styles with longer ground contact phase. According to Falbriard et al. [16], this maximum in the gyroscope signal can be detected reliably even though it has a speed-dependent bias from the actual TO. In their work, they provided speed-dependent biases for different fixed running speeds. Because the subjects in our study ran in wide speed bins of 1 m/s, we could not use the provided speed dependent biases. However, we used the authors’ overall 24 ms bias from the second maximum in the angular rate signal of the sagittal plane to find the index of the *i*-th TO event $n_{TO,i}$.

#### 2.3.2. Computation of Foot Trajectory

For the computation of the spatial parameters we implemented a strapdown integration algorithm to reconstruct the trajectory of the foot using a zero-velocity update to reduce IMU drift. The trajectory algorithm is an adapted version of the sensor fusion algorithm introduced by Rampp et al. [32]. It consists of a quaternion-based orientation estimation using the gyroscope data followed by a gravity corrected and dedrifted integration using the acceleration data.

The orientation estimation is based on the zero-velocity assumption. During MS the foot is assumed to be flat on the ground with a velocity of 0 m/s. Therefore, we can reset the orientation and the velocity for each stride during MS. For running, this assumption is often violated due to the dynamic nature of running for high running speeds. Besides, in the case of forefoot running no flat-foot phase exists. However, we could show in one of our prior publications [15] that the results for stride length and average stride velocity are still accurate using this assumption.

The following computational steps were applied to each running stride. Using the zero-velocity assumption, we initialized the orientation $\vec{\alpha }\left[n_{MS,i}\right]$ and the translation $\vec{s}\left[n_{MS,i}\right]$ of the foot at the *i*-th MS index $n_{MS,i}$ with zero:(3)$$\vec{\alpha }\left[n_{MS,i}\right]=\left(\begin{array}{c} 0\\ 0\\ 0 \end{array}\right)\;\vec{s}\left[n_{MS,i}\right]=\left(\begin{array}{c} 0\\ 0\\ 0 \end{array}\right)$$

After the initialization, we computed the orientation of the sensor during one stride using a quaternion-based forward integration. For this work, we use a vector representation for quaternions:(4)$$q=\left(\begin{array}{c} q_{0}\\ q_{1}\\ q_{2}\\ q_{3} \end{array}\right)$$

The quaternion q[n+1] at sample n+1 was computed from the previous quaternion q[n] for all samples between two consecutive MSs ($n\in [n_{MS,i};n_{MS,i+1}]$) using the following formulas [33,34].
(5)$$\begin{array}{c} q\left[n+1\right]=q\left[n\right]⊗dq_{\vec{\omega }\left[n\right]}=q\left[n\right]⊗\exp\left(\frac{1}{2f_{S}}W\left[n\right]\right)\;,\mathrm{with}\;W\left[n\right]=\left(\begin{array}{c} 0\\ \omega_{x}\left[n\right]\\ \omega_{y}\left[n\right]\\ \omega_{z}\left[n\right] \end{array}\right) \end{array}$$
(6)$$\exp\left(q\right)=exp\left(q_{0}\right)\left(\begin{array}{c} \cos\sqrt{q_{1}^{2}+q_{2}^{2}+q_{3}^{2}})\\ \frac{q_{1}}{\sqrt{q_{1}^{2}+q_{2}^{2}+q_{3}^{2}}}\sin\sqrt{q_{1}^{2}+q_{2}^{2}+q_{3}^{2}}\\ \frac{q_{2}}{\sqrt{q_{1}^{2}+q_{2}^{2}+q_{3}^{2}}}\sin\sqrt{q_{1}^{2}+q_{2}^{2}+q_{3}^{2}}\\ \frac{q_{3}}{\sqrt{q_{1}^{2}+q_{2}^{2}+q_{3}^{2}}}\sin\sqrt{q_{1}^{2}+q_{2}^{2}+q_{3}^{2}} \end{array}\right)$$

In Equation (Equation 5), the quaternion q[n] describes the rotation of the sensor from the initial position during MS $n_{MS}$ to the position at sample $n\in [n_{MS,i};n_{MS,i+1}]$. The quaternion q[n+1] is defined by rotating the quaternion q[n] by the differential quaternion $dq_{\vec{\omega }\left[n\right]}$ which describes the rotation during a time interval of duration $T=\frac{1}{f_{s}}$. This differential quaternion can be computed using the angular rate data $\vec{\omega }\left[n\right]$.

The quaternion sequence was used in two ways. First, the orientation of the sensor was computed in Euler angle representation by converting the quaternions to roll, pitch, and yaw:(7)$$\vec{\alpha }\left[n\right]=\left(\begin{array}{c} \mathrm{atan}\frac{2(q_{0}\left[n\right]q_{1}\left[n\right]+q_{2}\left[n\right]q_{3}\left[n\right])}{1-2(q_{1}{\left[n\right]}^{2}+q_{2}{\left[n\right]}^{2}])}\\ \mathrm{asin}(2\left(q_{0}\left[n\right]\cdot q_{2}\left[n\right]\right))\\ \mathrm{atan}\frac{2(q_{0}\left[n\right]q_{3}\left[n\right]+q_{1}\left[n\right]q_{2}\left[n\right])}{1-2(q_{2}{\left[n\right]}^{2}+q_{3}{\left[n\right]}^{2}])} \end{array}\right)$$

Second, the quaternion sequence was used for the gravity removal of the acceleration signal. The accelerometer constantly measures not only the acceleration of the foot movement, but also the gravitational acceleration. Using the determined quaternion sequence, the acceleration data $\vec{a}\left[n\right]$ were rotated from the shoe frame $\left(x_{s},y_{s},z_{s}\right)$ into the global frame $\left(x_{g},y_{g},z_{g}\right)$. The reason for this is that the sensor frame and the global frame coincide during MS and each quaternion q[n] describes the rotation from the sample at position *n* to the initial position at MS. In the global frame, the magnitude and direction of gravity was known, and thus we could remove it from the movement acceleration:(8)$$\begin{array}{c} \left(\begin{array}{c} 0\\ a_{x,gc}\left[n\right]\\ a_{y,gc}\left[n\right]\\ a_{z,gc}\left[n\right] \end{array}\right)=q\left[n\right]⊗\left(\begin{array}{c} 0\\ a_{x}\left[n\right]\\ a_{y}\left[n\right]\\ a_{z}\left[n\right] \end{array}\right)⊗q{\left[n\right]}^{-1}-\left(\begin{array}{c} 0\\ 0\\ 0\\ -9.81 \end{array}\right) \end{array}$$

As a last step, the translation of the sensor was computed by a dedrifted double integration of the gravity corrected acceleration signal ${\vec{a}}_{gc}\left[n\right]$. After the first integration of the acceleration signal over time, a linear dedrifting function was used to remove the velocity drift $\vec{\delta }\left[n\right]$ introduced by the integration of the acceleration. From the dedrifted velocity ${\vec{v}}_{dedrifted}\left[n\right]$, the translation $\vec{s}\left[n\right]$ of the sensor was computed by another integration over time:(9)$$\begin{array}{cc} \vec{v}\left[n\right] & =\sum_{m=0}^{n}\frac{1}{f_{s}}{\vec{a}}_{gc}\left[m\right]\\ \vec{\delta }\left[n\right] & =\frac{\vec{v}\left[n_{MS,i+1}\right]-\vec{v}\left[n_{MS,i}\right]}{n_{MS,i+1}-n_{MS,i}}\left(n-n_{MS,i}\right)\\ {\vec{v}}_{dedrifted}\left[n\right] & =\vec{v}\left[n\right]-\vec{\delta }\left[n\right]\\ \vec{s}\left[n\right] & =\sum_{m=0}^{n}\frac{1}{f_{s}}{\vec{v}}_{dedrifted}\left[m\right] \end{array}$$

The individual steps of the trajectory computation are visualized for one samples stride in Appendix A (Figure A1, Figure A2, Figure A3, and Figure A4).

#### 2.3.3. Parameter Computation

Based on the segmentation indices, the parameters stride time $t_{stride}$ and ground contact time $t_{gc}$ were computed as follows,
(10)$$\begin{array}{cc} t_{stride} & =\frac{n_{IC,i+1}-n_{IC,i}}{f_{s}}\\ t_{gc} & =\frac{n_{TO,i}-n_{IC,i}}{f_{s}} \end{array}$$

Stride length $d_{stride}$ and average stride velocity $v_{stride}$ were based on the translation of the foot obtained from the trajectory estimation:(11)$$\begin{array}{cc} d_{stride} & =\sqrt{s_{x}{\left[n_{MS,i+1}\right]}^{2}+s_{y}{\left[n_{MS,i+1}\right]}^{2}}\\ v_{stride} & =\frac{d_{stride}}{t_{stride}} \end{array}$$

Please note that we assumed level running and thus only use the *x* and *y* component of the translation due to the fact that the running path through the ground truth system was also flat. Thus, an error of the translation in *z*-direction was neglected.

In order to compute the angle parameters, one integration step was missing. We defined our stride from the *i*-th MS at $n_{MS,i}$ to the (i+1)-th MS at $n_{MS,i+1}$. For the computation of the sole angle of the *i*-th ground contact, the orientation in the sagittal between the *i*-th IC $n_{IC,i}$ and MS $n_{MS,i}$ was not computed. For the computation of the range of motion of the (i+1)-th ground contact phase, the last part of the eversion movement in the frontal plane happens after the (i+1)-th MS at index $n_{MS,i+1}$. Thus, we could neither compute both sole angle and range of motion for the *i*-th ground contact nor for the (i+1)-th ground contact. We decided to compute sole angle and range of motion for the first ground contact by adding a quaternion-based backward integration for the samples $n\in [n_{IC,i};n_{MS,i}]$.

We used Equation (Equation 5) for the computation of the backward integration. For this, the measured angular rate data was inverted by multiplying it by minus one. After that, the gyroscope values were integrated backwards from $n_{MS,i}$ to $n_{IC,i}$ by applying Equation (Equation 5). Finally, we converted the obtained quaternion sequence to the Euler angle representation using Equation (Equation 7) and concatenated it with the Euler angle orientation sequence obtained from $n_{MS,i}$ to $n_{MS,i+1}$.

In this orientation sequence, the sole angle is the angle obtained at the first IC $n_{IC,i}$ in the sagittal plane, because this angle describes the rotation of the shoe from IC to MS, where we assume the foot to be flat on the ground. Please note that a negative sole angle indicates a rearfoot runner, whereas a positive angle indicates a forefoot runner. The range of motion in the frontal plane is defined as the difference of the maximum and the minimum of the angle in the frontal plane between IC and TO (Figure 7).

### 2.4. Evaluation

To evaluate the effect of the sensor positions, we compared the raw IMU signals of the individual sensor positions with each other as well as the errors for the IMU-based spatio-temporal parameters.

#### 2.4.1. Evaluation of Raw Data Similarity

For the comparison of the raw signals, we used Pearson’s correlation coefficients [35]. We computed the correlation coefficients individually for each raw data axes between all sensor positions for the full stride ($n\in [n_{IC};n_{IC+1}]$), the ground contact phase ($n\in [n_{IC};n_{TO}]$), and the swing phase ($n\in [n_{TO};n_{IC+1}]$). We combined the correlation coefficients of the three acceleration and the three gyroscope axes for each sensor position pair and plotted the distribution of the correlation coefficients using boxplots. For the segmentation in ground contact phase and swing phase, we used the labels of the stride segmentation algorithm. In case the segmented signals had different length due to errors in the event detection, we cut the duration of all sensors signals to the shortest duration.

#### 2.4.2. Evaluation of Spatio-Temporal Parameters

To evaluate the spatio-temporal parameters with respect to the sensor positions, we computed the error of individual parameters for all sensor positions. For this work, we defined the error $E_{param}$ as follows,
(12)$$E_{param}=P_{sensor}-P_{gold}$$

In this formula, $P_{sensor}$ is the value of the parameter computed by the IMU sensor and $P_{gold}$ the value of the parameter determined by the ground truth. This formula indicates, that a positive error indicates an overestimation of the parameter and a negative error an underestimation of the parameter.

In order to understand the impact of running speed on the spatial parameters, we evaluated stride length $d_{stride}$ and the quality of the zero-velocity update for the different speed ranges. To evaluate the zero-velocity update, we computed the L2-norm of the difference of the acceleration signal at MS and the gravity vector in the global frame:(13)$$E_{\vec{a}}\left[n_{MS,i}\right]={∥\left(\begin{array}{c} a_{x}\left[n_{MS,i}\right]\\ a_{y}\left[n_{MS,i}\right]\\ a_{z}\left[n_{MS,i}\right] \end{array}\right)-\left(\begin{array}{c} 0\\ 0\\ 9.81 \end{array}\right)∥}_{2}$$

The idea of the error measure $E_{\vec{a}}\left[n_{MS,i}\right]$ is that no other acceleration except gravity should be measured by the accelerometer during MS $n_{MS,i}$. In case we also measure other accelerations, the zero-velocity assumption is violated and the error measure $E_{\vec{a}}\left[n_{MS}\right]$ increases.

## 3. Results

### 3.1. Results of Raw Data Similarity

Figure 8 visualizes the Pearson’s correlation coefficients as box plots. Each box represents the correlation coefficients of either the accelerometer or the gyroscope raw data in all three spatial directions between two sensor positions for all strides recorded during the data acquisition. We observe, that the sensors at different sensor position measure different signals, especially for the accelerometer signals. This means, that the sensor position has an impact on the IMU raw data. We can also see that the correlation coefficients for the cavity, heel, and lateral sensor position always yield the lowest values in combination with the instep sensor. Generally, the correlation coefficients of the raw data are higher for the gyroscope values as for the accelerometer values and the correlation coefficients are higher during swing phase than during ground contact phase.

### 3.2. Results of Spatio-Temporal Parameters

Table 3 lists the median errors and interquartile ranges (IQRs) of the IMU-based computation of stride time, ground contact time, stride length, average stride velocity, sole angle, and range of motion for the different sensor positions. We see that we can accurately measure stride time with all sensors. For ground contact time, we observe large IQRs of the errors for all sensor positions and higher median errors for the cavity and the instep sensor. For the sole angle, we observe higher median errors for the heel and the lateral sensor. For the range of motion in the frontal plane we find smaller errors than for the sole angle. For the parameters stride length and average stride velocity which are based on the computed translation, we see that the cavity sensor outperforms the other sensor positions with respect to both the median and the IQR.

Figure 9 depicts the error of the stride length $E_{d_{stride}}$ (Figure 9a) as well as the error of the acceleration at the zero-velocity updated $E_{\vec{a}}\left[n_{ms}\right]$ (Figure 9b) for the different sensor positions in the four speed ranges defined in our study (Table 2). We see that both the stride length as well as the error of the acceleration show larger errors for higher running velocities. Besides, the results are congruent with the median errors and show that the cavity sensor position outperforms the other sensor positions for the parameter stride length.

## 4. Discussion

Overall, no sensor position clearly outperforms all the other sensor positions even though the cavity sensor provides the best results for the translational parameters based on the reconstructed foot trajectory. In the following paragraphs, we will discuss the raw data comparison, the temporal parameters as well as the spatial parameters individually.

### 4.1. Differences in Raw Data

The correlation coefficients are higher for the gyroscope signals than for the acceleration signals (Figure 8). The reasons for this observation are twofold. On the one hand, the angular rate is less sensitive to movement artifacts than the accelerometer. Especially during the ground contact phase, the impact during IC introduces high frequency vibrations in the accelerometer signal, which differ for the individual sensor positions and cannot be correctly measured with a sampling frequency of $f_{s}$ = 200 Hz. On the other hand, the accelerometer measures different centripetal accelerations depending on the IMU position. The four main joints that cause the rotations in running are the hip, the knee, the ankle, and the metatarsal joint. Whereas the distance to the hip joint and the knee joint is similar for all sensors and thus neglectable with respect to differences in the centripetal accelerations, the raw signals differ with respect to the distance to the ankle and metatarsal joint. Thus, the accelerations measured by the IMUs will be different due to the distance dependent centripetal accelerations caused by the rotations around those joints. The rate of rotation is independent of the distance to the rotation center, if we assume the foot to be a solid object.

Additionally, the correlation coefficients during swing phase are higher than during ground contact phase (Figure 8). In particular, the gyroscope raw data of sensors at different positions can reach correlation coefficients close to one. During swing phase, the foot can be seen as a rigid segment, because there is no rotation around the metatarsal joint or any other distortion which might cause the sensors to rotate differently. The high correlation coefficients also indicate that the functional calibration procedure applied before the data acquisitions is capable of aligning the coordinate systems of the sensors at the different positions of the foot. If this was not the case, such high correlations would not be possible because rotations would be captured around different sensor axes. During ground contact phase however, the shoe upper is deformed due to deformation of the foot and the movement of the foot within the shoe. This causes movements at different positions on the shoe, which results in the measurement of different signals and lower correlation coefficients.

The correlation coefficients for the accelerations of the full strides show that the correlations to the instep sensors always yields the lowest values. We argue that the instep sensor is exposed to the highest motion by the deformation of the foot. During ground contact the arch of the foot and the forefoot flatten, which causes the upper of the running shoe to deform. This deformation of the upper also causes the laces and the tongue of the shoe to move and, consequently, the instep sensor as well.

More detailed analysis of the correlations reveals that the closer two sensors are located to each other, the higher their pairwise correlation. This makes sense, as we already discussed that the accelerations for the sensor positions differ depending on the distance to the joints causing the rotations. Thus, spatially close sensors have similar distances to those centers of rotation and and might be similarly effected by deformations of the shoe.

One limitation of our approach to comparing the raw signals is that the Pearson’s correlation coefficients might be influenced by differences in the detected events. Depending on the sensor positions, the events might be detected with a slight time shift. However, the high correlation coefficients during swing phase show, that the signals temporally match well and are not heavily influenced by time shifts.

In summary, the correlation coefficients show that the signals of the sensors at the four positions of the running shoe vary which causes different results for the spatio-temporal parameters. Especially the larger differences during ground contact have an effect on the accuracy of the parameters.

### 4.2. Temporal Parameters

The results for stride time $t_{stride}$ indicate that the bias corrected maximum of the angular rate signal in the sagittal plane is a reliable fiducial point which allows accurate estimations of the stride time for all sensor positions. With median errors of less than 0.5 ms and IQR of less than 8.6 ms (<2 samples at a sampling rate of 200 Hz) no sensor position clearly performs best. These accuracies for stride time are similar to the results presented by Falbriard et al. [16] who reported a stride time error 0±3 ms for their evaluation of stride time with an instep sensors. We explain the higher IQRs with the higher variability in speed in our data acquisition. Due to the fact that a sample-to-sample synchronization was not possible using our study setup, we could not evaluate the actual accuracy of the IC event for the different sensor positions. Thus, our results for stride time show only that we can reliably detect the fiducial point, but not that we can accurately detect IC with all sensor positions.

Generally, both the median errors and the IQRs for ground contact time $t_{gc}$ at all sensor positions are worse than for stride time $t_{stride}$. One of the reason for that is the inaccuracy of the TO detection. In running gait the IMU signal exhibits no clear feature at TO, whereas in walking gait, TO is indicated by a zero crossing of the angular rate data in the sagittal plane [32], the dynamic nature of the running gait does not exhibit such a feature [36]. Moreover, Falbriard et al. [16] showed that the bias of the local maximum in the sagittal plane used to detect TO is speed dependent. We neglected this speed dependency, because we had no information for the bias except for the fixed speeds that Falbriard et al. used in their study. We used the overall bias they reported to correct the index of the maximum in the sagittal plane which explains the high IQRs due to the high variance of speed in our study. The reasons for the higher median errors for the cavity and the instep sensor are not clear. It is possible that these results indicate an actual shift of the gyroscope maximum withing the gait cycle based on the sensor position. However, due to the lack of sample-to-sample synchronization, we were not able to further investigate this. In the future, a study with sample-to-sample synchronization could help to evaluate this assumption.

### 4.3. Spatial Parameters

For the orientation parameters, the cavity sensor positions shows the smallest median errors for both the sole angle and the range of motion. While the differences in accuracy between the sensor positions is smaller for the median error of the range of motion (2.3${}^{\circ }$ cavity/instep), the differences between the sensor positions is larger for the sole angle (7.7${}^{\circ }$ cavity/heel). The cavity and the instep sensors outperform the other two sensors with respect to the median error of the sole angle. We found the reason for these large differences in the bias we used to correct the local maximum in pitch angular velocity. When we removed the bias correction and used the local maximum as the IC event, we obtained different results (Table 4).

If no bias correction is applied during the event detection of IC the median errors are better for the heel and the lateral sensor position. Falbriard et al. [12] explained the high standard deviation for the sole angle evaluation in their work with the fact that the accuracy of the sole angle is heavily dependent on the accuracy of the IC detection algorithm. Due to the high angular velocities around IC, the integrated angle values are sensitive to the timing of the IC event. Based on the improved accuracy for a different bias for the heel and lateral sensor position we assume that the bias for the IC event is dependent on the sensor position itself. This underlines the need for a closer investigation of those biases for the different sensor positions. Nevertheless, we can conclude for the angle parameters, that the differences in the accuracy do not originate from the underlying raw data, but rather the stride segmentation algorithms.

For the stride length parameter the cavity position performs best, followed by the lateral position. The heel and the instep sensor position perform worst. These observations also hold for the different speed ranges (Figure 9a). We can see for all speed ranges that both the median errors and the IQRs are smaller for the cavity and the instep sensor, even though the accuracy drops with higher speed for those positions as well. One possible explanation for this fact is that the zero-velocity assumption during MS becomes less valid with higher speeds. The results for the error of the acceleration during zero-velocity phase are congruent with the errors in stride length (Figure 9b). The error $E_{\vec{a}}\left[n_{ms}\right]$ is smallest for the cavity sensor, followed by the lateral sensor, whereas the heel and instep sensor yield larger errors. Besides, the error increases with speed for all sensor positions. This shows that the zero velocity assumption is violated with higher speeds, but most valid for the cavity sensor. We argue that one of the reasons for this observation is the attachment of the sensors. The cavity inside the sole of the running shoes was manufactured to fit the shape of the miPod sensors and thus prevent the sensors from moving. The sole itself might act as a physical low pass filter by damping high frequency noise in the acceleration caused by impacts like IC by its elastic nature. Besides, the location of the sensor in the sole under the arch was chosen to be least affected by the deformation of the insole during IC and the bending in the forefoot region during the pushing phase. Thus, the cavity sensor position faces least perturbations during ground contact and can compute the zero-velocity-based trajectory with the highest accuracy. The heel and lateral sensor were taped to the heel cap and laterally under the ankle. While the attachment itself is also firm, those sensor positions are more affected by the movement of the upper of the shoe. Especially the heel sensor faces motion due to the deformation of the back of the running shoe caused by the heavy impact during IC for rearfoot runners. The instep sensor was mounted to the laces of the running shoe using a clip. Even though we tested the firmness of the attachment for each subject before the data acquisition, this sensor position is affected the most by additional movements. This is on the one hand due to the less firm attachment in comparison to the other sensor positions and on the other hand due to the highest amount of deformation of the upper of the running shoe, which we already discussed in the raw data section.

These movements and consequently the errors during MS harm the validity of the zero-velocity assumption which is the basis for the strapdown integration algorithm. Due to the fact that the sensor positions seem to be affected by additional movements or noise sources during MS to a different extent, the performance of the zero-velocity-based strapdown integration algorithm for the trajectory computation is different for the evaluated sensor positions.

The errors of the average stride velocity for the different sensor positions correlate with the ones for stride length. The reason for this correlation is that the average stride velocity is computed by dividing stride length by stride time. As the stride time computation works well for all sensor positions, the main error source for average stride velocity is the computed stride length. However, we can see from the small median errors and IQRs that average stride velocity can accurately be computed with foot worn IMUs.

### 4.4. General Aspects

Our results indicate that the raw signals and thus the results for the different spatio-temporal parameters differ for the evaluated sensor positions. This implicates that when comparing studies of IMU-based spatio-temporal parameters with each other, the sensor position should be considered as a source of difference in future review studies. Besides, adaptions to the algorithms like changing the detection algorithm of IC for the computation of the sole angle can result in performance boosts individual sensor positions. Thus, porting algorithms to other sensor positions can be possible, but algorithms might have to be adapted.

Nevertheless, we recommend the cavity sensor position due to the unobtrusive and firm attachment inside the running shoe sole. It is least affected by any additional movement than the actual foot movement and thus has the highest raw signal quality. However, we want to note at this point that this evaluation is purely based on the signal quality and the accuracy of the resulting spatio-temporal parameters and that we did not consider usability aspects. While the unobtrusiveness of the cavity sensor position is good for a smart shoe application with fully integrated sensors, it might not be a good solution for a sensor system which should be usable with different pairs of shoes. In this scenario, all the shoes would need a cavity and the sensors would need to be put under the sockliner before each run.

## 5. Conclusions

We presented an evaluation of the effects of four different IMU sensor positions on the accuracy of IMU-based endurance running parameters. We conducted a study with 29 subjects which were equipped with four IMUs placed inside a cavity in the sole of the running shoe, on the heel, the lateral side, and the instep of a running shoe. We compared the raw data of the individual sensor positions and implemented algorithms for stride segmentation and the computation of the trajectory of the foot, respectively, the running shoe. Using the data acquired during the study, we could show that the raw signals of the IMUs differ for the sensor positions, especially for the acceleration during the ground contact phase. We showed that all the sensor positions could accurately measure stride time, but not ground contact time due to the large speed dependency of the fiducial point we used to detect toe off. The angle parameters—range of motion and sole angle—were hard to compare for the different sensor positions, as they were affected by the accuracy of the IC event detection algorithm. Finally, we showed that the cavity sensor outperforms the other sensor positions for the computation of stride length, because this sensor positions seems not to be exposed to movement artifacts of the upper of the shoe.

Thus, we can conclude that the sensor position has an effect on the accuracy of different IMU-based running parameters due to the differences in the acquired raw signals. From a data processing perspective, we recommend to use IMU sensors inside the cavity of a running shoe even though it only outperforms the other sensor positions for the reconstruction of the trajectory using a zero-velocity-based strapdown integration algorithm.

In the future, the event detection algorithm for the different sensor positions should be investigated further. Because a sample-to-sample synchronization was not possible in our study set-up, we could not evaluate the accuracy of the actual IC/TO events but only the parameters deduced from those events. Our results indicate that, both for the IC and TO event, the biases from the maxima in the pitch angular velocity to the actual events are different for the different sensor positions. Improving the detection of the IC and TO event would not only increase the accuracy of ground contact time but also the stride angle parameters. Further, the larger errors for stride length with higher speeds should be further investigated. Especially for professional athletes, who run with speeds higher than 5 m/s, the errors in stride length are very high.

## Figures and Tables

**Figure 1 sensors-20-05705-f001:**
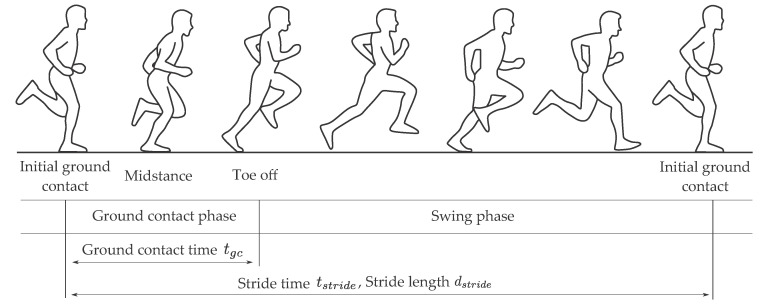
Visualization of the running gait cycle.

**Figure 2 sensors-20-05705-f002:**
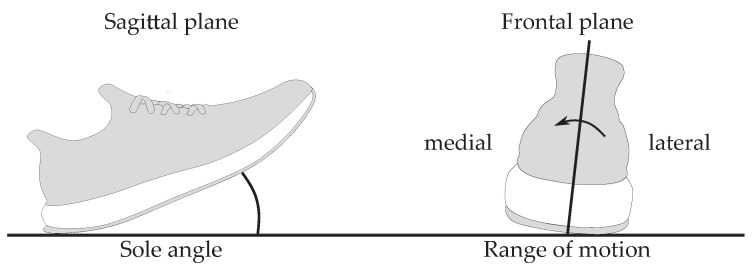
Visualization of sole angle and range of motion.

**Figure 3 sensors-20-05705-f003:**
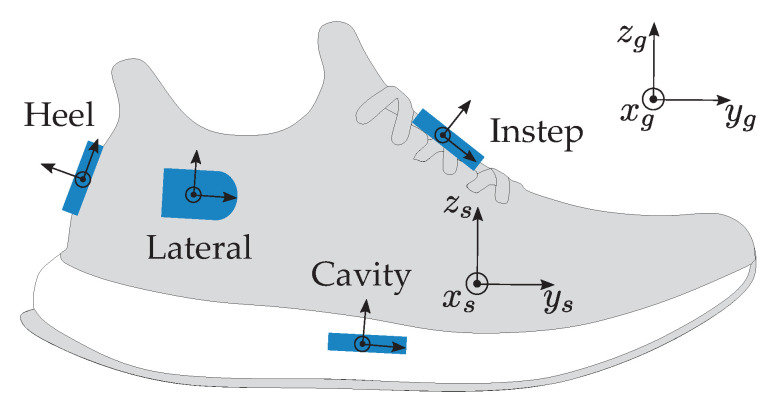
Visualization of sensor positions on the running shoes, the global coordinate system $\left(x_{g},y_{g},z_{g}\right)$, the shoe coordinate system $\left(x_{s},y_{s},z_{s}\right)$, and the individual sensor coordinate systems. When the foot is flat on the ground, the global and the shoe coordinate system are aligned.

**Figure 4 sensors-20-05705-f004:**
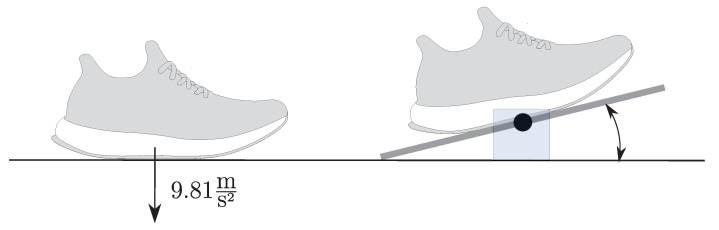
Visualization of the functional calibration procedure. The first part of the functional calibration consisted of standing still with the foot flat on the ground in order to measure gravity. During the second part the subjects rotated their feet on a balance board to compute the medial/lateral axis using a principal component analysis.

**Figure 5 sensors-20-05705-f005:**
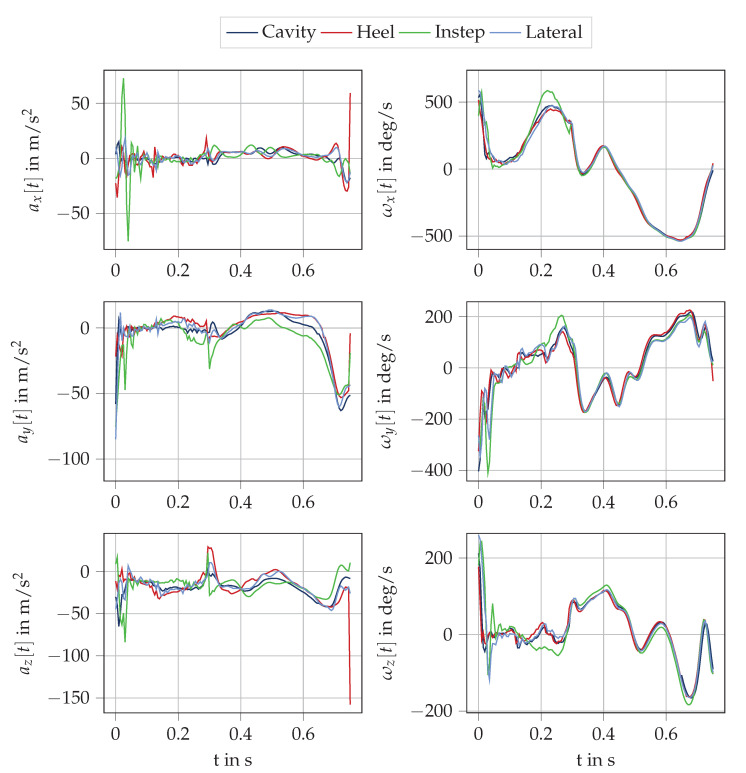
Exemplary IMU data of one stride segmented from IC to IC for the four different sensor positions.

**Figure 6 sensors-20-05705-f006:**
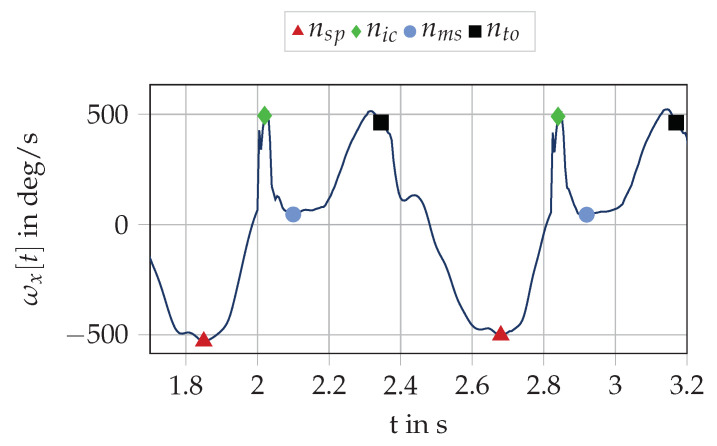
Visualization of the stride segmentation for the cavity sensor using the gyroscope signal in the sagittal plane $\omega_{x}\left[t\right]$. The fiducial points at swing phase $n_{SP}$ are local minima of the angular rate in the sagittal plane. The index $n_{IC}$ indicates the index of IC, which corresponds to the bias corrected local maximum after $n_{SP}$. The MS event $n_{MS}$ is at the minimum of the gyroscopic energy. The TO event at $n_{TO}$ is based on the second local maxima and a bias correction.

**Figure 7 sensors-20-05705-f007:**
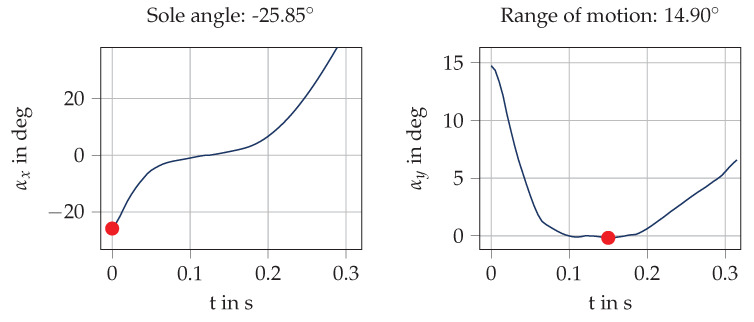
Visualization of angle computation for a sample stride from the cavity sensor. The angles are depicted from $n_{IC}$ (t=0 s) to $n_{TO}$ (t=0.32 s). The sole angle is defined as the rotation in the sagittal plane between IC and MS. As the orientation is initialized with zero at MS, the sole angle is the angle at $n_{IC}$. The range of motion is defined as the difference between the maximum and minimum (red dots) of the angle in the frontal plane during ground contact.

**Figure 8 sensors-20-05705-f008:**
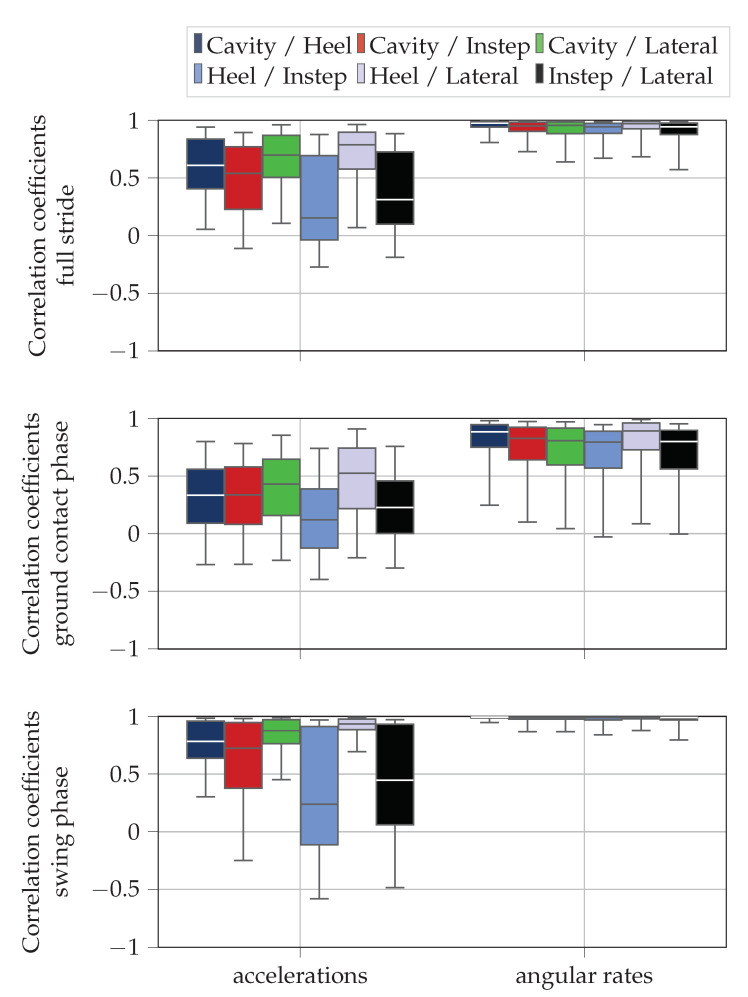
Results of the evaluation of the Pearson’s correlation coefficients between the IMU raw signals. Each box visualizes the correlation coefficients between two sensors for all the strides in *x*, *y*, and *z* direction. The box plots also display the median of the correlations (median line), the IQR (box), and the 5 and 95 percentiles (whiskers). The upper plot depicts the correlation of the full strides, the middle plot the correlations during the ground contact phase, and the lower plot the correlations during the swing phase.

**Figure 9 sensors-20-05705-f009:**
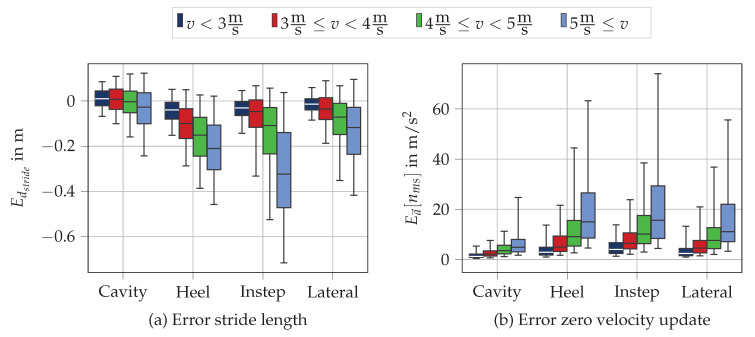
Visualization of the error for (**a**) stride length and (**b**) the acceleration at the zero-velocity update for the four different sensor positions in different speed ranges.

**Table 1 sensors-20-05705-t001:** Naming of the sensor position and details on the mounting of the sensors.

Name	Mounting
Cavity	Cavity cut in the sole of the shoe under the arch
Instep	Mounted with suiting clip to laces of the shoe
Lateral	Mounted with tape laterally under ankle
Heel	Mounted with tape on heel cap

**Table 2 sensors-20-05705-t002:** Number of trials and recorded strides per velocity range. During the data acquisition, we controlled for speed and the subjects only changed the velocity range, if the required number of trials in the previous (slower) velocity range was reached.

Velocity Range (m/s)	Number of Trials	Number of Strides
2–3	10	962
3–4	10	558
4–5	15	544
5–6	15	362

**Table 3 sensors-20-05705-t003:** Median error and IQR of the error for the parameters stride time, ground contact time, stride length, average stride velocity, sole angle, and range of motion compared to the motion capture system.

	Cavity		Heel		Instep		Lateral	
	Median	IQR	Median	IQR	Median	IQR	Median	IQR
Stride time (ms)	−0.5	6.9	0.0	8.4	0.4	7.6	0.3	8.6
Ground contact time (ms)	−11.0	37.6	−1.3	29.5	−22.6	37.5	−1.7	29.0
Sole angle (${}^{\circ }$)	1.6	7.2	−6.1	5.1	2.1	5.8	−5.9	5.1
Range of motion (${}^{\circ }$)	0.0	2.8	1.2	2.9	2.3	3.3	1.4	3.0
Stride length (cm)	0.3	8.5	−8.3	14.7	−5.6	15.1	−3.3	9.7
Avg. stride velocity (m/s)	0.0	0.1	−0.1	0.2	−0.1	0.2	0.0	0.1

**Table 4 sensors-20-05705-t004:** Median error and IQR of the error for the parameters sole angle without the bias correction for IC.

	Cavity		Heel		Instep		Lateral	
	Median	IQR	Median	IQR	Median	IQR	Median	IQR
Sole angle (${}^{\circ }$)	6.8	10.2	−2.9	6.8	6.7	7.0	−2.4	6.7

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
