# Peer review of "Does the Position of Foot-Mounted IMU Sensors Influence the Accuracy of Spatio-Temporal Parameters in Endurance Running?"

_sensors, 2020, doi:10.3390/s20195705_

Round 1

Reviewer 1 Report

This paper evaluates whether the sensor position of IMUs mounted to running
shoes has an impact on the accuracy of different spatio-temporal parameters. The paper is well written and easy to follow. The technical contributions are solid. I recommend the paper be accepted for publish after the following minor changes are made:

  1. Line 11 in abstract, "gold standard" do you mean groundtruth ?
  2. In Table 3, why are there negative values in the Table? Shouldn't all values measured be positive numbers?
  3. Figure 9, it would be good if the angle / heading accuracy is also evaluated.

Reviewer 2 Report

Introduction

The introduction session seems well organized, including the arguments that rationalize the study. I just would like to suggest changing the last paragraph, including only the aim of the study and some hypothesis.

Methods

Line 83 – Please consider writing in the past, not in the future.

The data analysis in the methods session is fully explained, allowing the replication of the study. However, information about the experiment is missing, such as, which was the running velocity and if it moved over time.

Discussion

The discussion is well written and very clear. My only questions were: is the accuracy of this measurement changing with the running velocity? could running velocity be a problem? In fact, the authors discuss in the second topic that running velocity but not related to the data accuracy.

Reviewer 3 Report

The paper approaches a study of the effects of the IMU sensor positions. The contribution is relevant in the scope of wearable technology. The manuscript is well-structured and clear. No changes are proposed.

Author Response

Answers Reviewer 3

Dear Reviewer

thank you for your positive feedback.

We hope that you also like the revised version of the manuscript including the suggestions of the other reviewers.

Best Regards
Markus Zrenner, Arne Küderle, Nils Roth, Ulf Jensen, Burkhard Dümler, Bjoern Eskofier

Reviewer 4 Report

This study primarily aimed to compare the effects of sensor position placement in the accuracy of running spatiotemporal parameters. 

First, I'd like to congratulate the authors for this detailed and in-depth analysis and description of the used methods. The use of several figures to illustrate the different steps are also very much appreciated. Well done and great job.

I very much appreciated the acknowledgment of the limitations when describing the methods/algorithms and the associated assumptions and potential bias. More importantly, how honest and transparent the authors were.

I have no comments on the introduction nor discussion as they look pretty clear and straightforward to me. I do have two suggestions regarding the data analysis approach:

  1. Instead (or as complement) of running pearson correlation, I believe cross correlation will provide you with more useful information. For example, the difference between cavity and heel might be a slight "time-shift" in the raw data that will certainly affect event detection. If that is the case, you can find higher correlation values but being able to report such "shift" at the temporal domain; this will tell the reader "the different positioning actually appear to measure the same but there is a timing issue to take into account"
  2. I know that filtering is a "giant world", but when dealing with this type of data, most of researchers filter the data to make it less noisy for event detection. I wonder if that wouldn't result in higher correlation values. 

While I understand that may not the main aim of the authors, I think these will generate useful information to report
